

# Exogenous melatonin improves germination rate in buckwheat under high temperature stress by regulating seed physiological and biochemical characteristics

Zemiao Tian[1,2], Mengyu Zhao[2], Junzhen Wang[3], Qian Yang[1], Yini Ma[1], Xinlei Yang[1], Luping Ma[1], Yongzhi Qi[1], Jinbo Li[4], Muriel Quinet[5], BaoSheng Shi[1] and Yu Meng[1]

[1] Hebei Agricultrual University, Baoding, China
[2] Chinese Academy of Agricultural Sciences, Institute of Crop Sciences, Beijing, China
[3] Liangshan Yi Autonomous Prefecture Academy of Agricultural Sciences, Xichang, China
[4] Luoyang Normal University, Luoyang, China
[5] Earth and Life Institute, Université Catholique de Louvain, Louvain-la-Neuve, Belgium

Corresponding authors
BaoSheng Shi, baoshengshi@163.com
Yu Meng, my131sohu@126.com

## ABSTRACT

The germinations of three common buckwheat (*Fagopyrum esculentum*) varieties and two Tartary buckwheat (*Fagopyrum tataricum*) varieties seeds are known to be affected by high temperature. However, little is known about the physiological mechanism affecting germination and the effect of melatonin (MT) on buckwheat seed germination under high temperature. This work studied the effects of exogenous MT on buckwheat seed germination under high temperature. MT was sprayed. The parameters, including growth, and physiological factors, were examined. The results showed that exogenous MT significantly increased the germination rate (GR), germination potential (GP), radicle length (RL), and fresh weight (FW) of these buckwheat seeds under high-temperature stress and enhanced the content of osmotic adjustment substances and enzyme activity. Comprehensive analysis revealed that under high-temperature stress during germination, antioxidant enzymes play a predominant role, while osmotic adjustment substances work synergistically to reduce the extent of damage to the membrane structure, serving as the primary key indicators for studying high-temperature resistance. Consequently, our results showed that MT had a positive protective effect on buckwheat seeds exposed to high temperature stress, providing a theoretical basis for improving the ability to adapt to high temperature environments.

## INTRODUCTION

Recent years have seen a marked increase in global temperature impacting the yield of various crops in numerous regions (*Mácová et al., 2022*). Predictions suggest that temperatures will continue to rise, underscoring the importance of research on crop

heat tolerance. High temperatures play a pivotal role in plant growth and development. However, exposure to excessively high temperatures can disrupt the cellular balance, alter cell homeostasis, and initiate membrane lipid peroxidation. These changes will ultimately inhibit plant growth (*Hao et al., 2021*). Seed germination, as the foundational phase of plant growth, is especially vulnerable to diverse stressors (*Penfield, 2017*). Exposure to temperatures beyond their optimal germination range primarily affects the seeds' morphological characteristics, compromising their development (*Ye et al., 2021*).

High-temperature stress can induce physiological and biochemical alterations in plant cells, resulting in membrane lipid oxidation and an overproduction of malondialdehyde (MDA). This sequence of events can damage cell membranes and may even lead to cell death (*Nahar et al., 2015*; *Roach, 2019*). Nevertheless, plants possess an intrinsic antioxidant defense system, regulating the activity of antioxidant enzymes, with the aim of mitigating oxidative damage. Elevating the activity of antioxidant enzymes, including peroxidase (POD), superoxide dismutase (SOD), and catalase (CAT), can decisively diminish plasma membrane damage and bolster tolerance to abiotic stress (*Ke et al., 2018*; *Alnusairi et al., 2021*; *Sajitha Rajan & Murugan, 2010*). In barley seeds research, it was observed that enzyme activity during germination at high temperatures was markedly elevated compared to that at moderate temperatures (*Mácová et al., 2022*). Additionally, *Sajitha Rajan & Murugan (2010)* posited a correlation between plant heat resistance and its inherent protective enzyme activity. At elevated temperatures, the generation rate of superoxide anions in cells increases substantially, surpassing the clearance rate of SOD, leading to reduced SOD activity that subsequently impacts seed germination. At the same time, the accumulation of osmoregulatory substances, such as proline, soluble sugars, and soluble proteins, regulates water potential, enhances cellular functions, and ensures a vigorous plant metabolism, thus increasing heat tolerance (*Zhang et al., 2015*). Research on corn has also indicated that elevating proline levels can increase its high-temperature tolerance (*Li, Ding & Du, 2013*). The utilization of exogenous plant growth regulators, such as osmoprotectants, and antioxidant compounds, is deemed a potent strategy to enhance plant stress resistance and yield.

Buckwheat, a staple crop across Asia and Europe, is an annual herbaceous plant classified under the Buckwheat genus of the Polygonaceae family (*Li et al., 2020*). It is rich in essential nutrients that are beneficial to human health, including vitamins, minerals, dietary fibers, proteins, and amino acids (*Alvarez-Jubete, Arendt & Gallagher, 2009*). Studies suggest that buckwheat thrives better than other crops in cooler climates (*Wen et al., 2021*). However, with the escalation of the greenhouse effect, temperatures in many buckwheat cultivation regions are on the rise. Given that buckwheat seeds have an optimal germination temperature of 22 °C, this thermal shift poses challenges to the germination process, necessitating immediate solutions. Although, some studies have reported the effects of high temperature on buckwheat Mosaic proteome, Mosaic hormone, and seedling growth (*Płazek et al., 2019*), there are few reports on buckwheat seed germination.

MT is a versatile growth regulator known to augment seed germination, and it has been shown to boost plant resilience against various unfavorable conditions (*Liang et al., 2018a*; *Liang et al., 2018b*; *Kaushal, 2020*). Extensive research indicates that MT's pivotal

role in seed germination and plant growth may be attributed to alterations in physiological mechanisms influenced by MT. Specifically, MT fosters regular seed germination by amplifying the activity of antioxidant enzymes (*Galano, Tan & Reiter, 2011*). As a broad-spectrum antioxidant, MT can enhance the activity of antioxidant enzymes, such as SOD and POD, which protects the plants from stress-induced damage (*Gao et al., 2018*; *Wang, 2018*. Previous results have shown that the appropriate concentration of MT can reduce the MDA content in soybean seedlings, alleviating the damage to the membrane system and improving resistance to abiotic stress (*Wei, 2015*). Treatment with exogenous MT significantly increased the content of osmotic regulators in plants, while increasing cell fluid concentrations and reducing the MDA content in plants under abiotic stress (*Kaushal, 2020*). Moreover, MT seed treatments can refine the antioxidant system and starch metabolism under cold stress, further facilitating seed germination (*Marta, Szafrańska & Posmyk, 2016*; *Cao et al., 2019*). In addition, the effects of MT on buckwheat under drought stress have been reported. MT can promote stress, improve resistance and protective activity, reduce accumulation of reactive oxygen species, and increase substances (*Li et al., 2020*).

Currently, how MT modulates the germination of buckwheat seeds and its physiological mechanisms of resistance to high temperature stress is still a subject to be explored. Thus, the objectives of this study are to (1) investigate the impact of high temperature stress and then the positive effect of MT soaking on the morphology and physiology of buckwheat seeds; (2) investigate whether the antioxidant enzyme system plays a major role in reducing the damage of high temperature stress on seed germination in response to MT seed soaking.

# MATERIALS & METHODS

## Plant material and reagents

This experiment was conducted in the laboratory of the College of Landscape and Tourism, Hebei Agricultural University. The test materials consisted of five buckwheat varieties Fenghuangtehong ('Fhth'), Kanbaotianqiao ('Kbtq'), Zhongku NO. 3 ('Zk3h'), Chuanqiao NO. 8 ('Cq8h'), and Xinong 9978 ('Xn9978') The buckwheat seeds were sourced from China Agriculture Provided by the Buckwheat Gene Resources Innovation Research Group of the Institute of Crop Science, Chinese Academy of Sciences. MT, with a purity of 99%, was purchased from Sigma Company in the United States (Burlington, MA, USA).

## Determination of high temperature resistance

Buckwheat seeds were surface sterilized with 75% ethanol for 30 min and then rinsed with distilled water five times (*Bai et al., 2020*). Sterilized seeds were placed in Petri dishes (30 seeds and five replicates per petri dish) with three layers of filter paper and distilled water. Seeds in petri dish were germinated in incubators (the light intensity was 4,000 LX and the photoperiod was 8–10 h) at different temperatures of 22, 25, 28, 30 °C. To determine the suitable high temperature, GR and GP were measured.

## Determination of MT concentration

Before this test, we first selected 1,500 uniformly sized and plump seeds, soaked in 75% ethanol for 30 min, followed by rinsing (with distilled water) and drying. We then treated these seeds in a dark place for 16 h with MT at different concentrations (0, 50, 100, 200, 500 uM). We selected 30 seeds for each concentration, placed them in a petri dish, and repeated this process five times. We recorded the number of seed germinations daily, calculated the germination potential and germination rate.

## Experimental setup

In subsequent tests, seeds were soaked in 200 uM MT at 22 °C for 16 h using distilled water as a control, and then germinated in the same way as above. Approximately 10 g of embryo and radicle were collected from each treatment at 2, 4 and 6 days after germination and then stored at −80 °C for subsequent analysis (superoxide dismutase, peroxidase activity, and osmotic regulation). There were four treatments in total: (1) water immersion + 22 °C (CK); (2) water immersion + 28 °C (H); (3) 200 μM MT immersion + 22 °C (MT); (4) 200 μM MT immersion + 28 °C (MT+H).

## Measurement of seed germination parameters

Germination was standardized by the length of the radicle when it reached half the length of the seed. The number of germinations was recorded from day 1 to day 7 of seed germination. GP and GR were calculated on the 3rd and 7th days after germination (*Li, Yu & Yin, 2017*). The formulas are as follows:

$$GP = \frac{\text{number of seeds germinated on the 3rd day}}{\text{number of seeds tested}} \times 100\%$$

$$GR = \frac{\text{number of seeds germinated on the 7th day}}{\text{number of seeds tested}} \times 100\%.$$

## Measurement of growth indicators

After seven days of germination of seeds, 30 seeds were taken from each treatment. The radicle length was measured using a vernier caliper. This process was repeated three times for each treatment (*Li, Yu & Yin, 2017*; *Thabet et al., 2018*). The fresh weight of the seeds was measured using an electronic balance.

## Measurement of physiological and biochemical indicators

The method described by *Sarker (2019)* with slight modification was used to estimate soluble sugars (Ss) content. Briefly, 0.3 g seed sample was added to 9 mL of distilled water in a boiling water bath for 30 min. A total of 1 ml of the supernatant was mixed with 5 mL of sulfuric acid-anthrone reagent, then boiled for 10 min and cooled. The absorption value was measured in a spectrophotometer at 620 nm.

The proline (Pro) content was estimated following the method described by (*Subramanyam, Du & Van, 2019*). Briefly, approximately 0.3 g frozen sample was homogenized with three mL of 3% aqueous sulfosalicylic acid. After centrifugation, 2 mL

of supernatant was mixed with 2 mL of glacial acetic acid and 2 mL of ninhydrin reagent. The solution was then heated in boiling water for 30 min. After cooling, the solution was centrifuged at 10,000 rpm for 5 min. The light absorption values were recorded at 520 nm using a spectrophotometer.

The soluble protein (Sp) content was determined according to the Coomassie brilliant blue method described by *Jiang et al. (2018)*. To prepare the Coomassie Brilliant Blue G-250 solution, 0.1 g of the dye was dissolved in 50 ml of 90% ethanol. This was followed by the addition of 100 ml of 85% $H_3PO_4$. The final volume was adjusted to 1,000 ml with distilled water, and the solution was stored in an amber glass bottle for one month. For the assay, a precise volume of 0.1 ml of the enzyme solution was mixed with 0.9 ml of distilled water, followed by the addition of 5 ml of the prepared Coomassie Brilliant Blue G-250 reagent. The mixture was thoroughly homogenized and allowed to stand for 2 min. Absorbance readings were then taken at a wavelength of 595 nm.

Relative electrical conductivity (REC) was assessed using a conductivity meter, in accordance with the method described by *Lim et al. (2022)*. Fresh ornamental buckwheat samples were initially rinsed with deionized water and dried to remove surface moisture. The samples were then accurately weighed following cutting, and 0.1 g of each sample was placed in centrifuge tubes (in triplicate). These samples were soaked in deionized water for 24 h. Conductivity measurements were first taken (R1), and subsequently, after cooling the samples in boiling water for 20 min, a second measurement (R2) was conducted.

Catalase (CAT) activity was quantified using the method described by *Bałabusta, Szafránska & Posmyk (2014)*. For each assay, a 50 ml Erlenmeyer flask was used (the second as the control), The test flask received 2.5 ml of enzyme extract, while the control flask contained 2.5 ml of boiled enzyme extract. To each, 2.5 ml of $H_2O_2$ was added. The mixtures were then incubated in a constant temperature water bath at 30 °C for 10 min. Subsequently, 2.5 ml of 10% $H_2SO_4$ was added to terminate the reaction. Titrated with 0.1 mol/L $KMnO_4$ until pink (30s not gone).

Malondialdehyde (MDA) was determined by the acid ninhydrin method (*Landi, 2017*). A 0.1 g sample of the tissue was weighed and homogenized with 2 ml of 10% trichloroacetic acid (TCA) and a small amount of quartz sand. An additional 8 ml 10% TCA was added to the homogenate, which was then transferred to a 10 ml centrifuge tube and centrifuged at 2,000 rpm for 10 min. The supernatant served as the sample extract. For the assay, 3 ml of 0.5% thiobarbituric acid (TBA) and 1 ml of the enzyme extract were mixed in a centrifuge tube. The mixture was then incubated in a boiling water bath for 30 min, cooled immediately, and centrifuged at 9,000 rpm for 15 min. The absorbance of the supernatant was measured at 450 nm, 532 nm, 6,00 nm wavelength to determine the absorbance value, take TBA as blank.

The reaction mixture contained the following: 1.5 ml of pH 7.8 phosphate buffer, 0.3 ml of methionine (MET), 0.3 ml of NBT, 0.3 ml of EDTA-Na, 0.25 ml of distilled water, 0.05 ml of enzyme extract, and 0.3 ml of riboflavin. Control tubes contained the same mixture, replacing the enzyme extract with phosphate buffer. One control tube was kept in the dark while the others were exposed to 4,000 LX (25 °C light incubator, level 6 light) for 20 min. The reaction was terminated by covering the tubes with black plastic bags. The

absorbance of the samples was measured at 560 nm, using the unilluminated tube as a blank.

Superoxide dismutase (SOD) activity was evaluated using the nitroblue tetrazolium (NBT) reduction method (*Stephenie et al., 2020*). The assay involved two sets of test tubes: one for treatment and the other for sample analysis. The reaction mixture contained the following: 1.5 ml of pH 7.8 phosphate buffer (only for test tubes), 0.05 ml of enzyme solution (only for control), 0.3 ml of methionine (MET), 0.3 ml of NBT, 20.3 ml of EDTA-Na, 0.25 ml of distilled water, 0.05 ml of enzyme solution, 0.3 ml of riboflavin. Two test tubes were used pH 7.8 phosphate buffer instead of enzyme solution. For the two test tubes, one tube was placed in the dark, while the other was reacted in 4,000 LX (25 °C light incubator, level 6 light) sunlight for 20 min. After the reaction was over, they were covered with black plastic bags to terminate the reaction. The absorbance of the samples was measured at 560 nm, using the unilluminated tube as a blank.

Peroxidase (POD) was determined using the guaiacol method (*Stephenie et al., 2020*). The reaction mixture consisted of 2.9 ml phosphate buffer, 1 ml of 2% $H_2O_2$ 1 ml of guaiacol solution, and 0.1 ml of enzyme solution (enzyme solution was added at the end). The mixture was placed in a 37 °C water bath for 15 min, followed by rapid cooling in an ice bath. To terminate the reaction, 2 ml trichloroacetic acid was added. Absorbance was measured at 470 nm, using a control consisting of 2.5 ml of heated and boiled enzyme liquid.

## Statistical analysis

Excel was used to record data, and SPSS 26.0 (IBM Corp., Armonk, NY, USA) was used to perform statistical analysis. Duncan's test was used to compare average values and to conduct the lowest significant difference test. We also conducted correlation analysis, path analysis, and principal component analysis. Prism 9.0 Statistical software (https://www.graphpad-prism.cn) was used for chart production. Tbtools (https://bio.tools/tbtools) and Chiplot online (https://www.chiplot.online/) were used for cluster analysis.

(1) Correlation analysis (*Adler & Parmryd, 2010*).

Correlation analysis (Pearson correlation) measures the degree of correlation between two variables. The value range is from −1 to 1. A value of 1 indicates a complete positive correlation, −1 indicates a complete negative correlation, and 0 indicates there is no linear correlation.

The relevant calculation formula is:

$$R = \sum (X_i - X_{av}) \times (Y_i - Y_{av}) \div \sqrt{\sum (X_i - X_{av})^2 \times \sum (Y_i - Y_{av})^2}.$$

where $X_i$, $Y_i$ represents different values of the measurement index, $X_{av}$, $Y_{av}$ represents the corresponding average number of the measured indicators.

(2) Path analysis (*Lu et al., 2022*).

This study uses SOD, POD, CAT, Sp, Ss, Pro, MDA as independent variables, and REC as the dependent variable (Y). We first conducted a Y normality test. Then, a stepwise regression analysis was conducted to obtain the linear regression equation. Based on the direct path coefficient and linear regression equation obtained by stepwise regression

**Table 1 The daily germination rate of seeds treated with different temperature.**

| Temperature (°C) | GP (%) | GR (%) |
|---|---|---|
| 22 | 34 ± 6.41 a | 42 ± 3.8 a |
| 25 | 32 ± 2.98 a | 39.3 ± 2.78 a |
| 28 | 20 ± 2.36 b | 25.3 ± 1.83 b |
| 30 | 1.3 ± 1.83 c | 1.3 ± 1.83 c |

Notes.

Different lowercase letters indicate significant differences at the 0.05 probability level ($P < 0.05$).

analysis, the indirect path coefficient (n) of the respective variables to the dependent variable is calculated. The related calculations formula is:

$$n = r_{ij} \times P_{jy}.$$

In the formula, $r_{ij}$ is the correlation coefficient between $X_i$ and $X_j$; $P_{jy}$ is the direct path coefficient of $X_j$; $P_i$ is the direct path coefficient of the independent variable.

(3) Membership function analysis.

The method of *Schroeter-Zakrzewska & Pradita (2021)* was used to calculate the membership function value ($U_m$) of each measurement indicator. The calculation formula is:

$$U_m = (C_{im} - C_{imin}) \div (C_{imax} - C_{imin})$$

where $C_{imax}$ represents the maximum value of the mth measurement index; $C_{imin}$ represents the minimum value of the mth measurement index; the greater the average value of the membership function, the stronger the high temperature resistance of the variety.

# RESULTS

## Selection of temperature

The data presented in Table 1 reveals a notable inverse relationship between temperature and both the GP and GR of buckwheat. Specifically, a comparative analysis against the control condition (22 °C) highlighted a decrement in GP of buckwheat at elevated temperatures. At a temperature of 25 °C, there was a marginal decrease in germination potential by 5.88%, which was statistically insignificant. However, at temperatures of 28 °C and 30 °C, the reductions in germination potential were substantial, registering decreases of 41.18% and 96.18%, respectively, compared to the control.

Similarly, GR of buckwheat exhibited a decline of 6.43% at 25 °C, which progressively worsened to 39.76% at 28 °C and 96.9% at 30 °C, in comparison to the control. The reductions at 28 °C and 30 °C were statistically significant. It was observed that at 30 °C, buckwheat seeds exhibited pronounced stress symptoms, indicating a severe impact on germination. In contrast, at 28 °C, the stress on seed germination was moderate. This response pattern aligns with the preference of ornamental buckwheat for a warm and cool growth environment. Consequently, the optimal temperature for seed germination was identified as 22 °C. For the purposes of investigating the effects of high-temperature stress, 28 °C was selected as the critical temperature threshold.

**Table 2 Effects of different melatonin concentrations on germination potential and germination rate of buckwheat seeds under high temperature stress.** Different lowercase letters indicate significant differences at the 0.05 probability level ($P < 0.05$).

| MT concentration ($\mu$mol/L) | GP (%) | GR (%) |
|---|---|---|
| 0 | 18.67 ± 4.47 bc | 24 ± 4.35 bc |
| 50 | 20.67 ± 2.79 ab | 25.33 ± 2.98 abc |
| 100 | 23.33 ± 3.33 ab | 28 ± 3.8 ab |
| 200 | 26 ± 4.35 a | 31.33 ± 5.06 a |
| 500 | 14.67 ± 3.8 c | 20.67 ± 4.35 c |

GR of buckwheat decreased by 6.43, 39.76, and 96.9% compared with the control at 25, 28, and 30 °C, respectively. The reductions at 28 °C and 30 °C were statistically significant. It was observed that at 30 °C, buckwheat seeds exhibited pronounced stress symptoms, indicating a severe impact on germination. In contrast, at 28 °C, the stress on seed germination was moderate. This response pattern aligns with the preference of ornamental buckwheat for a warm and cool growth environment. Consequently, the optimal temperature for seed germination was identified as 22 °C. For the purposes of investigating the effects of high-temperature stress, 28 °C was selected as the critical temperature threshold.

## Selection of MT concentrations for seed treatment

Upon establishing 28 °C as the critical high-temperature stress condition for buckwheat, a study was conducted to evaluate the effects of varying concentrations of MT on seed germination of 'Kbtq'. As detailed in Table 2, both GP and GR of buckwheat exhibited an initial increase followed by a decrease correlating with rising concentrations of MT.

Treatment groups with 50, 100, and 200 μM MT exhibited an increase in germination potential by 10.71, 24.96, and 39.26%, respectively, compared to the control group. Notably, the enhancement in the 200 μM MT group was statistically significant. Conversely, seeds treated with 500 μM MT demonstrated a reduction in GP by 21.46% compared to water-soaked seeds. Furthermore, GR was observed to increase by 5.54%, 16.67%, and 30.54% for the 50, 100, and 200 μM MT treatments, respectively, with the 200 μM concentration showing a significant enhancement relative to water-soaked control seeds. In contrast, the 500 μM MT treatment resulted in a 13.88% decline in GR, and a marked decrease of 34.02% was noted for the 200 μM treatment compared to water-soaked seeds. Our results suggest that MT concentrations at 50, 100, and 200 μM positively influence buckwheat seed germination, whereas higher concentrations (500 μM) appear to inhibit it. The optimal exogenous MT concentration for alleviating high temperature stress in ornamental buckwheat was identified as 200 μM, which had the most pronounced effect on enhancing seed germination.
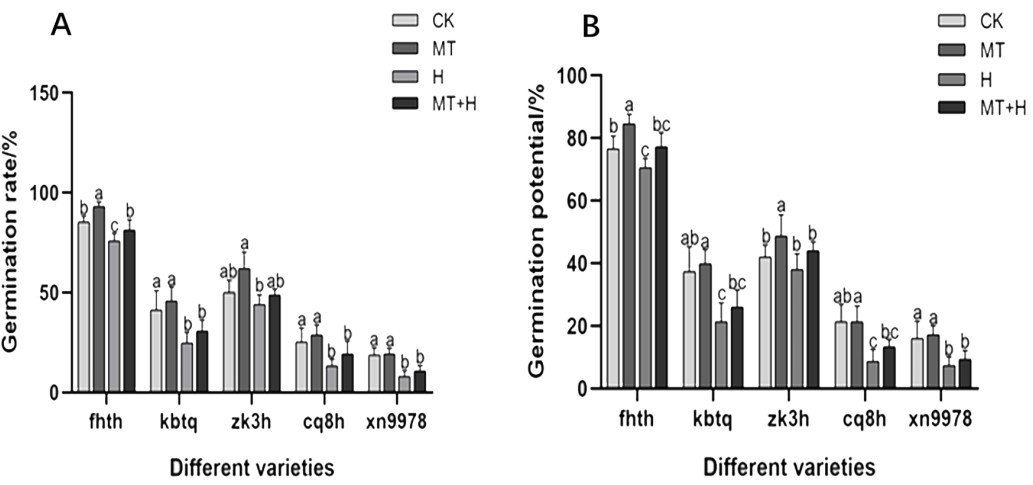

**Figure 1** Effect of MT on germination rate (A) and germination potential (B) of different buckwheat seeds under high temperature stress.

## Impact of exogenous MT on buckwheat seed germination under high temperature stress

In this research, we assessed the impact of MT treatment on the GP of five different buckwheat varieties. Among these, 'Fhth' exhibited a notable increase of 10.43% in GP under MT treatment compared to CK treatment. A consistent trend was observed in seed GR and GP across treatments. Specifically, under MT treatment, the germination rates of 'Fhth' and 'Zk3h' increased by 9.38 and 24%, respectively, relative to CK treatment. By the second day, variations in germination conditions across the five buckwheat varieties became evident, highlighting the effectiveness of MT. The H treatment adversely affected seed germination, leading to diminished vigor and rate. Except for 'Zk3h', the other four varieties ('Fhth', 'Kbtq', 'Cq8h', and 'Xn9978') displayed considerably lower GR under H treatment compared to CK treatment. Under MT+H treatment, GPs of 'Fhth', 'Kbtq', 'Zk3h', 'Cq8h', and 'Xn9978' increased by 9.43, 21.88, 15.79, 53.85, and 27.27% respectively in comparison to the H treatment (Fig. 1). GRs of 'Fhth', 'Kbtq', 'Zk3h', 'Cq8h', and 'Xn9978' under the MT+H treatment were increased by 7.02, 24.32, 10.6, 45 and 33.33% respectively compared with the H treatment. These results suggest that MT treatment enhances buckwheat seed germination, with 'Fhth' and 'Zk3h' benefiting most prominently. Likewise, the MT+H treatment significantly increases germination under H conditions, especially for varieties 'Kbtq' and 'Zk3h', with 'Cq8h' exhibiting an exceptionally notable enhancement.

## Impact of exogenous MT on growth indicators of buckwheat seeds under high temperature stress

Under MT treatment, the radicle lengths of 'Fhth' and 'Xn9978' saw substantial increases of 27.63% and 9.11% respectively compared to CK treatment. Conversely, the H treatment curtailed both the radicle length and fresh weight across all five buckwheat varieties relative to CK treatment. Specifically, the radicle length of 'Kbtq', 'Cq8h', and 'Xn9978'

**Table 3  Influence of melatonin treatment on radicle length and fresh weight of different varieties of buckwheat seeds under high temperature stress.** Germination outcomes for various varieties across four treatments.

| Varieties | Treatment | Radicle length (cm) | Fresh weight (g) |
|---|---|---|---|
| 'Fhth' | CK | 4.70 ± 0.31 bc | 0.36 ± 0.02 b |
| | MT | 6.00 ± 0.36 a | 0.43 ± 0.03 a |
| | H | 4.28 ± 0.46 c | 0.28 ± 0.02 c |
| | MT+H | 4.97 ± 0.41 b | 0.31 ± 0.03 c |
| 'Kbtq' | CK | 3.71 ± 0.32 a | 0.39 ± 0.02 b |
| | MT | 3.99 ± 0.22 a | 0.45 ± 0.03 a |
| | H | 3.04 ± 0.29 b | 0.32 ± 0.03 c |
| | MT+H | 3.71 ± 0.28 a | 0.38 ± 0.02 b |
| 'Zk3h' | CK | 2.89 ± 0.25 ab | 0.23 ± 0.02 b |
| | MT | 3.20 ± 0.21 a | 0.27 ± 0.02 a |
| | H | 2.51 ± 0.24 b | 0.15 ± 0.02 c |
| | MT+H | 2.81 ± 0.40 ab | 0.21 ± 0.01 b |
| 'Cq8h' | CK | 3.74 ± 0.22 ab | 0.22 ± 0.02 b |
| | MT | 3.93 ± 0.21 a | 0.30 ± 0.02 a |
| | H | 3.18 ± 0.34 c | 0.15 ± 0.01 c |
| | MT+H | 3.43 ± 0.42 bc | 0.20 ± 0.02 b |
| 'Xn9978' | CK | 3.51 ± 0.19 b | 0.37 ± 0.02 b |
| | MT | 3.83 ± 0.16 a | 0.43 ± 0.02 a |
| | H | 3.10 ± 0.24 c | 0.29 ± 0.02 c |
| | MT+H | 3.59 ± 0.22 ab | 0.34 ± 0.01 b |

**Notes.**

CK, water treatment; MT, 200 $\mu$mol/L MT treatment; H, 28 °C treatment; MT+H, 200 $\mu$mol/L MT+28 °C treatment. 'Fhth', 'Kbtq', 'Xn9978':three sweet buckwheat varieties, 'Zk3h' and 'Cq8h': two varieties of tartary buckwheat. Different lowercase letters indicate significant differences at the 0.05 probability level ($P < 0.05$). Error bars represent standard errors of three repeated calculations.

underwent significant reductions, and the fresh weight of all five varieties decreased markedly. However, when subjected to MT+H treatment, radicle lengths of the five buckwheat varieties grew by 16.02, 22.17, 11.86, 7.93 and 15.88% respectively against the H treatment, with 'Fhth', 'Kbtq', and 'Xn9978' registering significant increments. Furthermore, seed fresh weights under MT+H treatment rose by 12.62, 17.14, 33.94, 28.7 and 16.94% respectively relative to the H treatment. With the exception of 'Fhth', MT treatment markedly augmented the fresh weight in the other four varieties. In summary, exogenous MT mitigated the adverse effects of high temperature stress on buckwheat radicle growth and fresh weight (Table 3).

## Effect of exogenous MT on Pro in buckwheat seeds under high temperature stress

The Pro content in 'Cq8h' seeds increased significantly by 22.94% and 19.77% on days 2 and 4 respectively with MT treatment, when compared to the CK treatment. Similarly, for 'Fhth' and 'Zk3h' seeds on days 2, 4, and 6 under MT treatment, the Pro content showed a significant increase by 31.47, 33.11, 16.81% and 14.87, 28.69, 20.83% respectively compared to the CK treatment. The H treatment led to a noticeable rise in the intraspecific

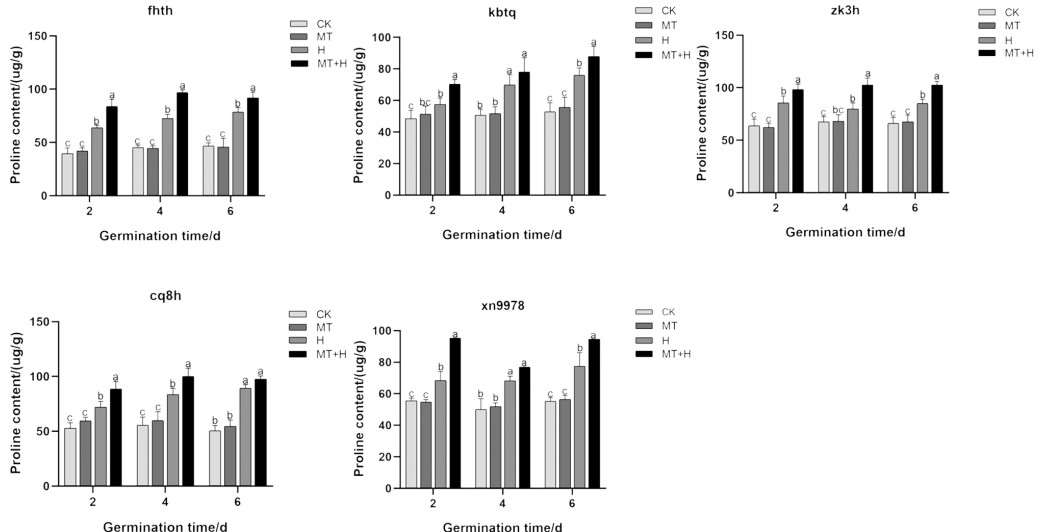

**Figure 2** **Effect of MT on Pro in various buckwheat seed varieties under high temperature stress.**

Pro content of buckwheat in comparison to the CK treatment. Notably, 'Fhth' and 'Cq8h' seeds exhibited a pronounced proline accumulation, suggesting enhanced stress resistance. When considering 'Kbtq' and 'Xn9978' seeds under MT+H treatment on days 2 and 6 post-germination, there was a significant Pro content rise by 22.12, 15.52, 37.23 and 31.09%, respectively compared to the H treatment. 'Cq8h' seeds under MT+H treatment only on days 2 and 4 after germination registered a marked increase in Pro content by 22.94 and 19.77% respectively when compared to the H treatment. Lastly, the other two varieties, on days 2, 4, and 6 under MT+H treatment, recorded a significant Pro content surge compared to the H treatment, with increases of 31.47, 33.11, 16.81% for 'Fhth' and 14.87, 28.69, 20.83% for 'Zk3h'. Overall, the data suggest that exogenous MT can elevate the Pro content in buckwheat seeds subjected to high temperature stress (Fig. 2).

## Effect of exogenous MT on Ss in buckwheat seeds under high temperature stress

On the 2nd day post-germination, the Ss content in 'Fhth' seeds under MT treatment showed a significant increase of 19.57% compared to the CK treatment. On day 4 for 'Xn9978', a notable rise of 14.91% in Ss content was observed under MT treatment when juxtaposed with the CK treatment. Additionally, the Ss content of the other three varieties under MT treatment showed a marked increase on days 4 and 6, with increments of 19.91, 64.33% for 'Kbtq', 23.71, 24.63% for 'Zk3h', and 17.44, 23.73% for 'Cq8h' respectively compared to the CK treatment. The H treatment caused a distinct surge in Ss content, particularly in 'Zk3h', on days 4 and 6. Other varieties also displayed a significant increase in Ss content on days 4 and 6 under H treatment when compared to the CK treatment. Comparing MT+H to H treatment on days 2, 4, and 6, the Ss content in 'Fhth', 'Kbtq', 'Zk3h', and 'Cq8h' rose significantly with values of 16.6, 28.39, 49.31% for 'Fhth', 17.57, 16.65, 36.02% for 'Kbtq', 25.84, 32.41, 25.18% for 'Zk3h', and 35.12, 22.14, 45.01% for

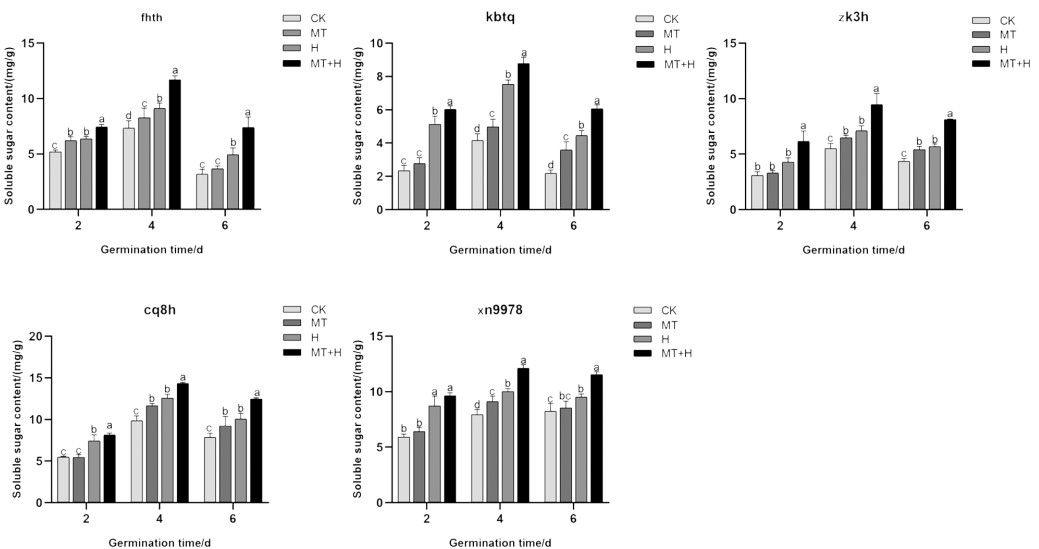

**Figure 3** Effects of MT on Ss content in various buckwheat seed varieties under high temperature stress.

'Cq8h' (Fig. 3). The findings demonstrate that MT treatment can boost Ss content in buckwheat seeds, and this enhancement is more pronounced under the MT+H treatment, thereby helping mitigate the detrimental effects of high temperature on seeds.

## Effects of exogenous MT on Sp of buckwheat seeds under high temperature stress

The Sp content in the five varieties subjected to MT treatment was notably higher on the 4th and 6th days of seed germination compared to those under CK treatment, though the difference was negligible on the 2nd day (Fig. 4). In comparison to the CK treatment, H treatment significantly enhanced the soluble protein content across the five varieties. As germination progressed, the fluctuations in Sp content mirrored those of Ss content. Excluding 'Fhth', which exhibited a consistent increase, the Sp content in the other four varieties followed an initial increase followed by a decrease. On days 2, 4, and 6 of germination, the Sp levels under the MT+H treatment surpassed those of the H treatment, with increases of 21.74, 16.72, 24.15% for 'Fhth' and 15.02, 12.87, 18.56% for 'Zk3h'. For the remaining varieties, only on days 4 and 6, did the MT+H treatment result in significant rises in Sp content, with increases of 16.17, 14.85% ('Kbtq'), 15.49, 20.39% ('Cq8h'), and 8.68, 13.65% ('Xn9978'). The findings suggest that MT effectively boosts Sp content.

## Effect of exogenous MT on the SOD enzyme activity of buckwheat seeds under high temperature stress

Post MT treatment, the SOD activity in the five varieties surpassed that of the CK treatment. Specifically, 'Kbtq' demonstrated a notable rise on the 6th germination day, 'Zk3h' on the 4th day, and 'Xn9978' on the 2nd, 4th, and 6th days. H treatment led to an uptick in SOD activity. When contrasted with the Ck treatment, SOD activity rose significantly on days 2,

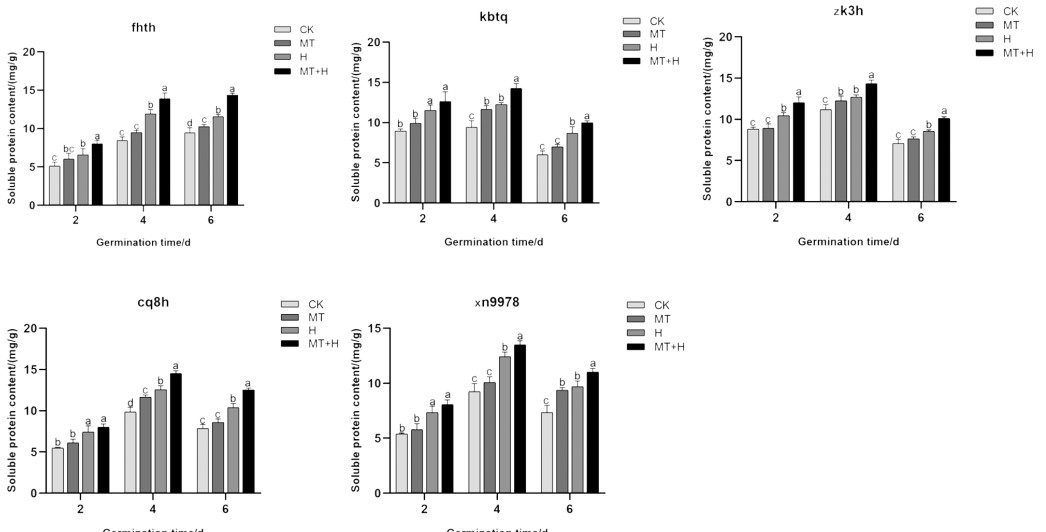

**Figure 4** Impact of MT on Sp of different buckwheat seeds under high temperature stress.

4, and 6 post germination. Generally, SOD activity across treatments exhibited a pattern of initial augmentation followed by a decrease, peaking on the 4th germination day. Relative to the H treatment, MT+H treatment resulted in significant SOD activity increases of 11.62, 16.11, 11.81% for 'Fhth', 11.21, 11.77, 9.85% for 'Kbtq', and 9.35, 5.48, 6.97% for 'Xn9978' on days 2, 4, and 6 post germination. 'Zk3h' displayed SOD activity boosts of 19.69, 19.54, and 13.21% under the MT+H compared to the H treatment, with significant enhancements on days 2 and 4. 'Cq8h' exhibited SOD activity growths of 19.74, 9.37 and 22.83% under MT+H compared to H treatment, notably on days 2 and 6 (Fig. 5). These findings illustrate that SOD activity across the five buckwheat varieties heightened under H and MT treatments relative to CK.

## Influence of exogenous MT on POD enzyme activity in buckwheat seeds under high temperature stress

The trend in POD activity largely mirrored that of SOD activity, with POD activity exhibiting a notable increase across all treatments on the 4th day. Specifically, the POD activity of 'Kbtq', 'Zk3h', and 'Cq8h' markedly rose under MT treatment in comparison to the CK treatment on days 2, 4, and 6 germinations. The 'Fhth' variety showed a significant increase on day 2, while 'Xn9978' experienced a pronounced rise on the 4th germination day. H treatment elevated POD activity across the buckwheat varieties, but the MT+H treatment yielded even higher POD activity than the H treatment. The trends in POD activity for 'Kbtq' and 'Fhth' were analogous, with both varieties exhibiting similar patterns on days 2 and 4. Under the MT+H treatment, POD activity significantly exceeded that of the H treatment, increasing by 41.88, 25, 22.54, and 19.61% respectively. This differential was not marked on the 6th germination day, with 'Zk3h' only showing an uptick on this day. POD activity under the MT+H treatment for 'Zk3h' and 'Cq8h' significantly increased compared to the H treatment, by 24.53 and 24.06% respectively on day 4. For 'Xn9978',

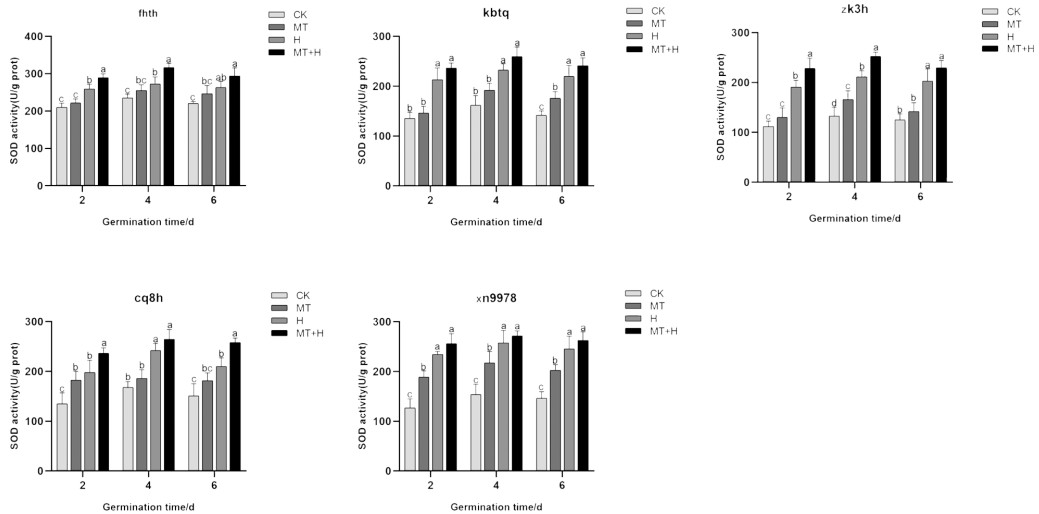

**Figure 5 Impact of MT on SOD enzyme activity of different buckwheat seeds under high temperature stress.**

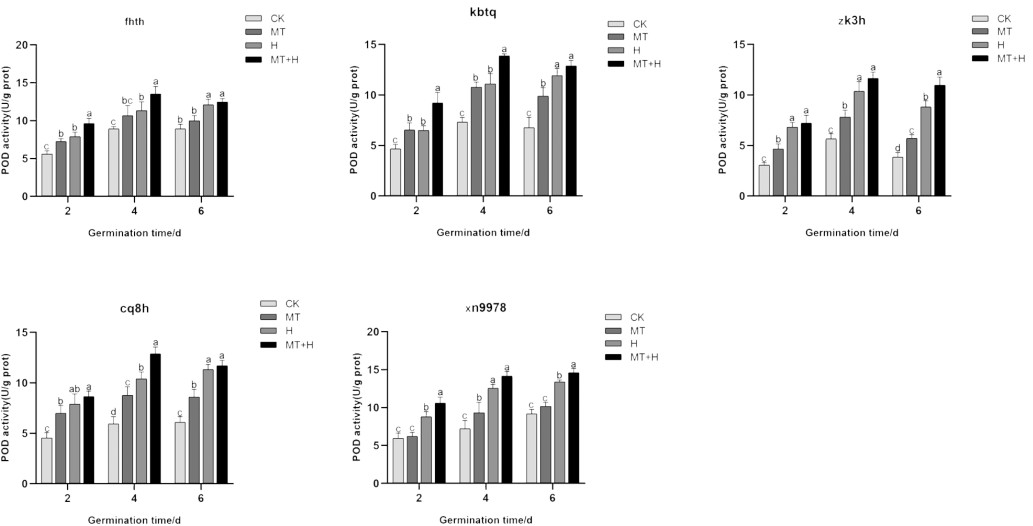

**Figure 6 Impact of MT on POD enzyme activity of different buckwheat seeds under high temperature stress.**

on days 2 and 6 post germination, the MT+H treatment led to POD activity rises of 20.88 and 9.13% relative to the H treatment, with no notable change on the 4th day (Fig. 6). These observations suggest that MT enhances the POD activity in the five buckwheat seed varieties, and its presence appears to mitigate the detrimental effects on buckwheat seed germination under high temperature stress by amplifying POD activity.

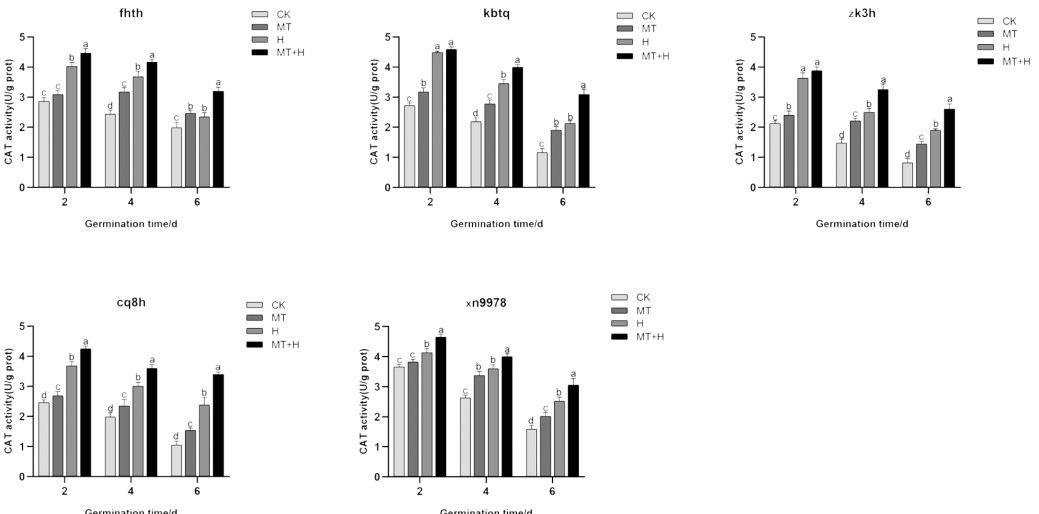

**Figure 7** Influence of MT on CAT enzyme activity of different buckwheat seeds under high temperature stress.

## Effect of exogenous MT on CAT enzyme activity in buckwheat seeds under high temperature stress

Contrary to the SOD and POD activities, CAT activity peaked on 2nd day, subsequently declining as germination time extended. When compared to the CK treatment, the MT treatment significantly augmented the CAT activity across all buckwheat varieties. For 'Fhth' and 'Xn9978' on the 4th and 6th germination days, CAT activity under the MT treatment rose significantly by 30.23, 24.29, 27.95, and 26.79% compared to the CK treatment. Furthermore, the CAT activities of 'Kbtq', 'Zk3h', and 'Cq8h' in the MT treatment showed significant enhancements at days 2, 4, and 6 post germination relative to the CK treatment. CAT activities across the five buckwheat varieties were significantly elevated under the H treatment compared to the CK treatment. The CAT activity trends for 'Zk3h' and 'Kbtq' were analogous. On days 4 and 6, the CAT activity from the MT+H treatment notably surpassed that of the H treatment, with increases of 15.57, 46.67, 30.68, and 37.31%, though this distinction was insignificant at 2 days post germination (Fig. 7). Experimental outcomes indicate that a 200 µM MT treatment substantially boosts the CAT activity in buckwheat seeds.

## Influence of exogenous MT on MDA content in buckwheat seeds under high temperature stress

The MDA content across five buckwheat varieties post MT treatment was diminished compared to the CK treatment. Specifically, 'Fhth' recorded significant reductions on days 2 and 4, 'Kbtq' on day 6, and 'Zk3h' on days 4 and 6, while 'Cq8h' showed a notable decrease on day 4. Although 'Xn9978' also exhibited a decline, this change wasn't statistically significant. Over the course of germination, the MDA content in the five varieties subjected to H treatment exhibited an ascending trend, with the most substantial accumulation observed on day 6. Employing a 200 µM MT treatment effectively curtailed

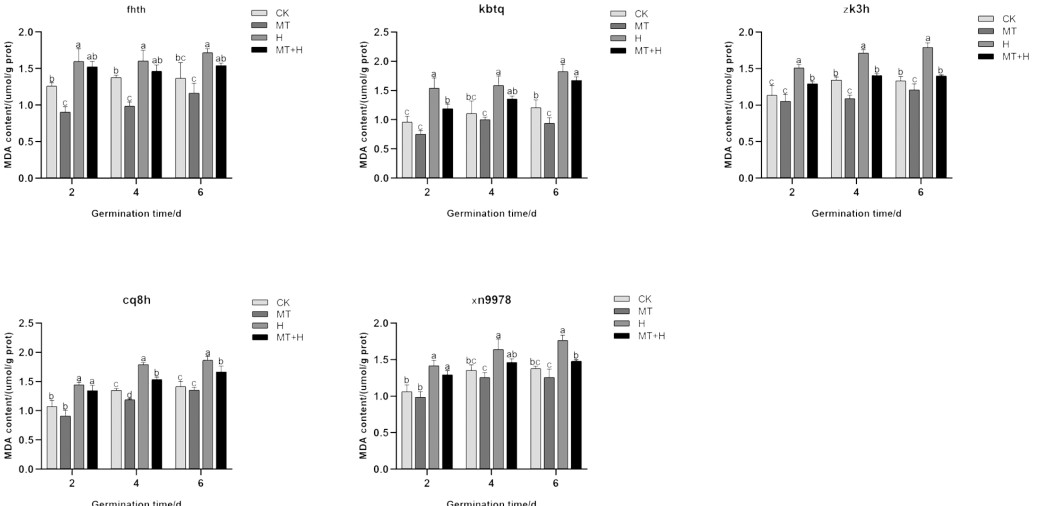

**Figure 8  Effect of MT on the MDA content of different buckwheat seeds under high temperature stress.**

MDA accumulation within these varieties. On days 2, 4, and 6 of germination, the MDA content under the MT+H treatment was decreased relative to the H treatment across the varieties. MDA levels in 'Fhth' and 'Zk3h' seeds were significantly reduced by 9.02 and 8.71% (on day 2), 10.85 and 13% (on day 4), 16.88 and 21.39% (on day 6), respectively. On the 2nd day of germination of 'Kbtq', the MDA content of MT+H treatment was lower than H treatment. The MDA content of the H treatment was significantly reduced by 20.05%; on the 4th and 6th days of 'Cq8h', the MDA content of the MT+H treatment was significantly reduced by 12.51 and 11.56% compared with the MDA content of the H treatment; 'Xn9978' was on the 6th day, the MDA content in the MT+H treatment was significantly reduced by 15.34% compared with the MDA content in the H treatment (Fig. 8). The test results show that 200 μM MT treatment can reduce MDA accumulation in both normal and high-temperature environments, effectively alleviating the damage caused by membrane lipid peroxidation.

## Correlation analysis and principal component analysis based on each measurement index

Correlation analysis elucidated the relationships among various indices under high temperature stress and MT treatment. Of the seven physiological parameters, the relationship between SOD and POD stands out as particularly significant, registering a correlation coefficient of 0.82. Robust correlations exist between SOD and CAT, POD and Ss, and POD and Sp, while the links between Ss, Sp, and MDA are comparatively weaker (Fig. 9). The effects of MT under high temperature stress were evaluated by performing principal component analysis on morphological and physiological indicators of all treatments. The contribution rates of the two principal components PC1 and PC2 extracted by the principal component analysis were 57.5% and 30.9% (PC1), 57.1%, and 14% (PC2), respectively. The various indicators are clearly distinguished according to PC1

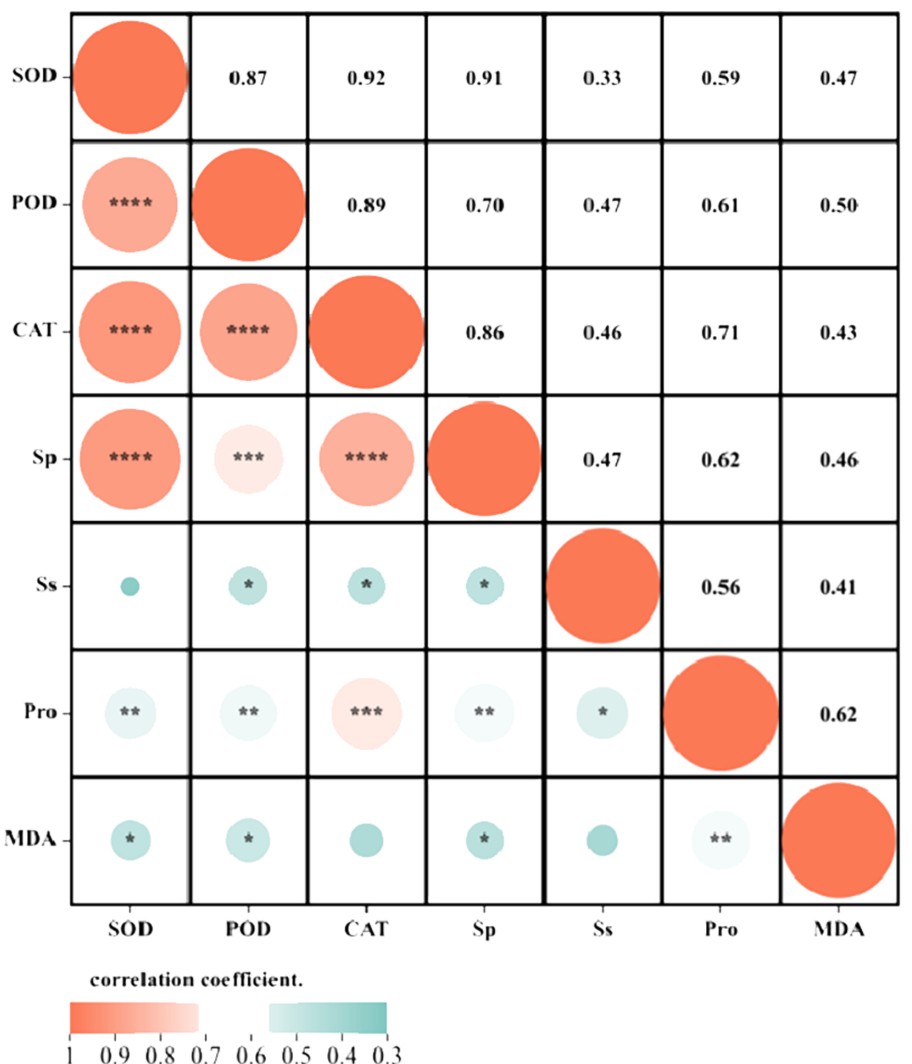

**Figure 9  Correlation analysis of the correlation between physiological indicators.**

and PC2. The results show that among the morphological indicators, GP, GR, and RL have the largest eigenvalues and have a large contribution to PC1 and PC2. Fresh weight has a large contribution to PC1, while REC has a great contribution to PC2 (Fig. 10A). From the physiological parameters, SOD and POD possess the highest eigenvalues, significantly impacting both PC1 and PC2. Pro, Ss, and Sp mainly contribute to PC2, with MDA having the least influence on both PC1 and PC2 (Fig. 10B).

## Gray correlation analysis of exogenous MT on buckwheat seed germination under high temperature stress

The correlation between seed physiological indicators and MT seed soaking concentration under high temperature stress is shown in Fig. 11. The correlation between POD activity, Sp content, and MT concentration is the most relevant. These indicators can be used to measure

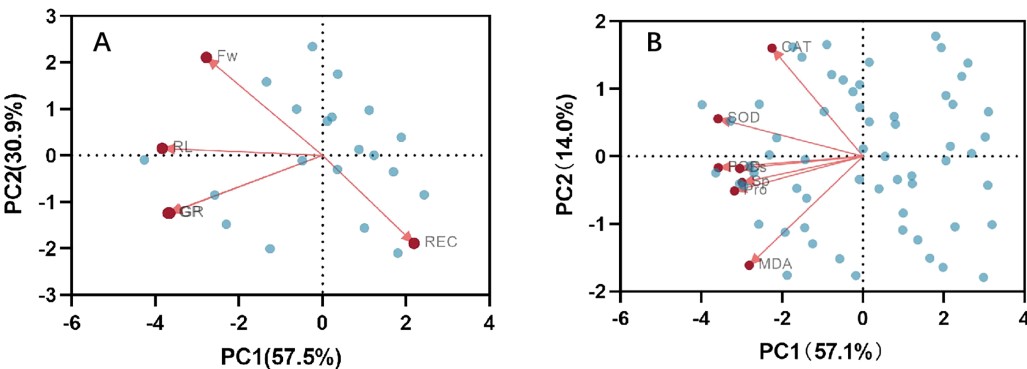

**Figure 10  Principal component analysis index of (A) morphology and (B) physiological correlation.**

the alleviating effect of MT on buckwheat seed germination under high temperature stress. The results show that POD activity and Sp content are the most comprehensive indicators for evaluating the effect of MT in alleviating high temperature stress. In addition, Ss, Pro, MDA content, SOD activity, and CAT activity also significantly reflected the alleviating effect of MT on buckwheat seed germination under high temperature stress.

## Effect of exogenous MT on the relative conductivity of buckwheat seeds under high temperature stress

Under standard temperature conditions, the relative conductivity of 'Kbtq' and 'Xn9978' seeds treated with MT was notably lower than those treated with CK, decreasing by 15.56% and 10.78%, respectively. The relative conductivity of the other three varieties did not change significantly. The relative conductivity of the five varieties of buckwheat seeds increased with increasing temperature, and the relative conductivity of the H treatment increased significantly compared with the CK treatment. The MT treatment significantly reduced the relative conductivity of buckwheat seeds under the H treatment. 'Cq8h' and 'Xn9978' were reduced by 15.78% and 13.27% respectively compared with the H treatment (Table 4). Trials indicate that a 200 μM MT treatment can notably reduce seed conductivity and bolster cell membrane resilience against stress-induced damage.

## Path coefficient analysis of MT on physiological indicators of buckwheat seeds under high temperature stress

The magnitude of the path coefficient delineates the influence of physiological indicators on buckwheat seeds' membrane permeability under combined treatments. Concurrently, the magnitude of the indirect path coefficient reflects the potential damage extent to the cell membrane as induced by these physiological indicators. Path analysis outcomes highlight SOD, POD, CAT, Sp, Ss, Pro, MDA, and REC as pivotal factors. SOD exhibits the most pronounced effect on REC with a path coefficient of 1.2, whereas Sp exerts the least influence, evidenced by its 0.005 coefficient (Fig. 12). Notably, POD, CAT, and MDA profoundly affect REC *via* the mediation of SOD, Sp, Ss, and Pro. Among these, each

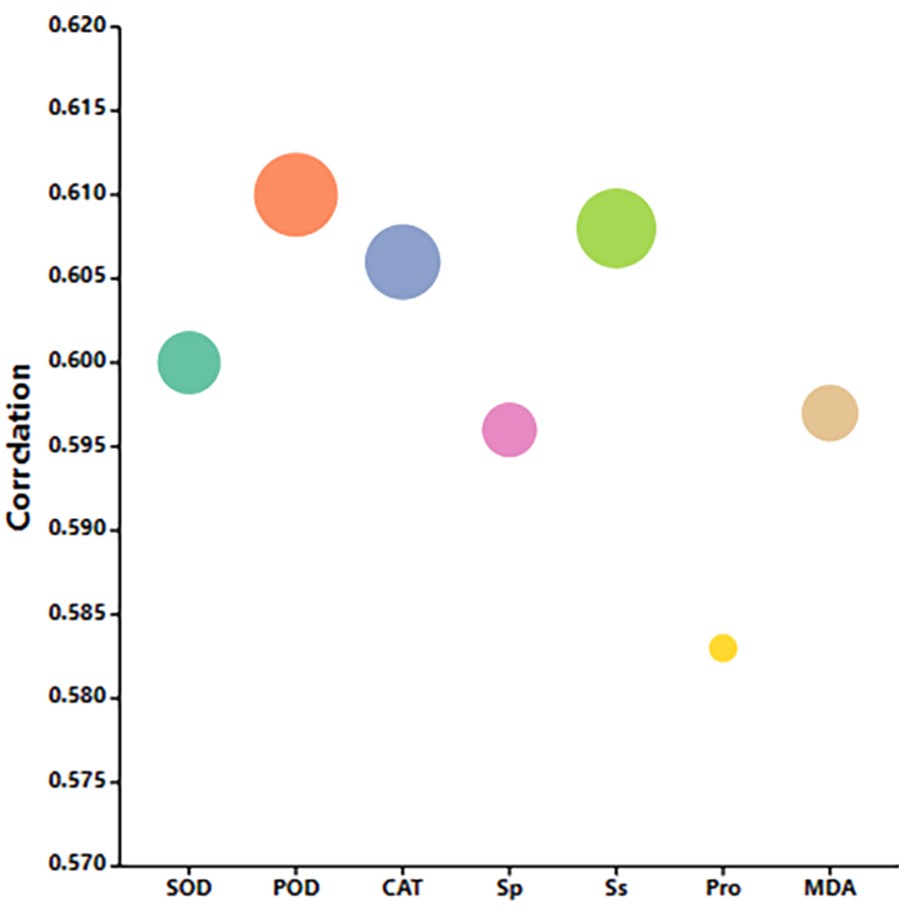

**Figure 11  Analysis of the grey relational grade of MT and high temperature stress.**

**Table 4  Effect of melatonin treatment on the relative conductivity of buckwheat seeds under high temperature stress (relative conductivity of different varieties under four treatments).**

| Varieties | Relative electrical conductivity(%) | | | |
|-----------|------|------|------|------|
|           | CK   | MT   | H    | MT+H |
| 'Fhth'    | 55.24 ± 3.9 bc | 49.16 ± 6.09 c | 74.08 ± 3.62 a | 62.39 ± 7.17 b |
| 'Kbtq'    | 50.59 ± 4.02 c | 42.72 ± 2.55 d | 70.02 ± 3.48 a | 60.73 ± 3.4 b |
| 'Zk3h'    | 49.52 ± 2.83 c | 47.78 ± 3.63 c | 80.73 ± 2.98 a | 72.25 ± 5.0 b |
| 'Cq8h'    | 53.46 ± 6.75 c | 48.95 ± 4.66 c | 77.37 ± 3.06 a | 66.24 ± 4.0 b |
| 'Xn9978'  | 43.58 ± 3.06 c | 38.88 ± 1.49 d | 68.76 ± 3.63 a | 62.11 ± 1.84 b |

**Notes.**
CK, water treatment; MT, 200 μmol/L MT treatment; H, 28 °C treatment; MT+H, 200 μmol/L MT+28 °C treatment. 'Fhth', 'Kbtq', 'Xn9978': three sweet buckwheat varieties, 'Zk3h' , 'Cq8h': two varieties of tartary buckwheat. Different lower-case letters indicate significant differences at the 0.05 probability level ($P < 0.05$). Error bars denote standard errors of three replicate calculations.

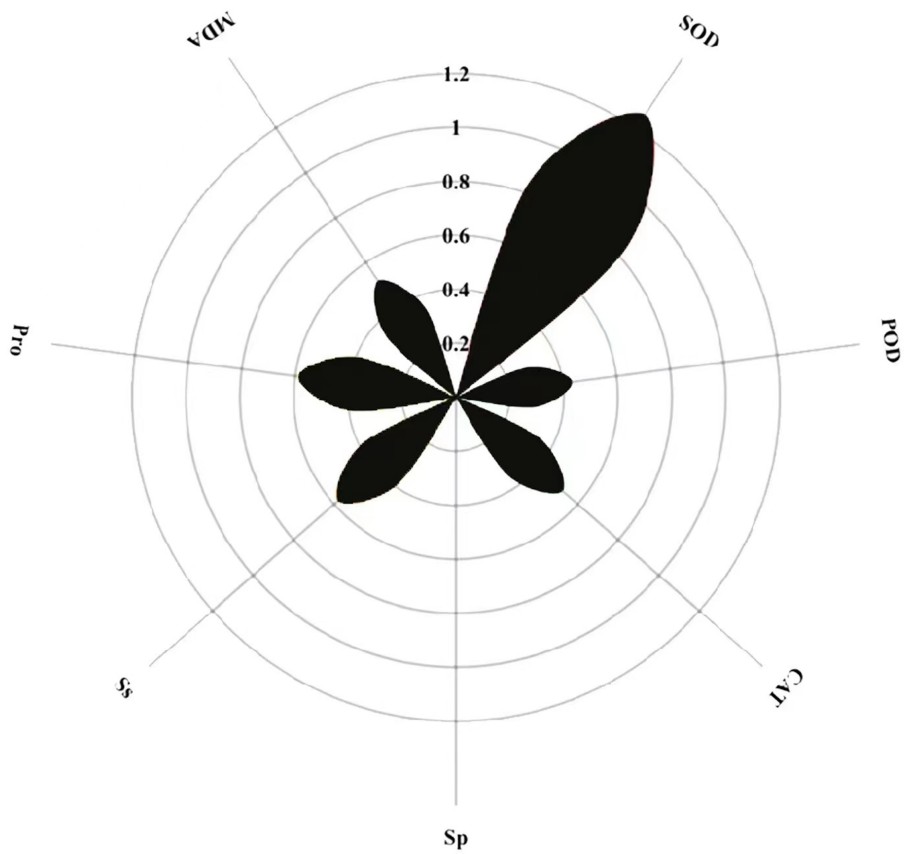

**Figure 12** **Direct path analysis of REC (relative conductivity) and physiological indicators.**

metric's indirect impact on REC is most accentuated through SOD, then Ss, with MDA having the least indirect influence.

## Comprehensive evaluation of exogenous MT on buckwheat seed germination tolerance under high temperature stress

The membership function method was used to conduct a comprehensive evaluation of buckwheat under high temperature treatment, MT treatment, and combined treatment (Fig. 13). The results show that under H treatment, the membership function value of the measurement index of 'Fhth' is significantly higher than the membership function value of other varieties, indicating that 'Fhth' has the strongest high temperature resistance. Under MT treatment, the membership function value of 'Fhth' is the largest, indicating that MT has a significant effect on 'Fhth'. Under the combined treatment of high temperature and MT, the membership function values of 'Fhth' are the best high, and the membership function value of 'Kbtq' is higher than that of 'Zk3h'.

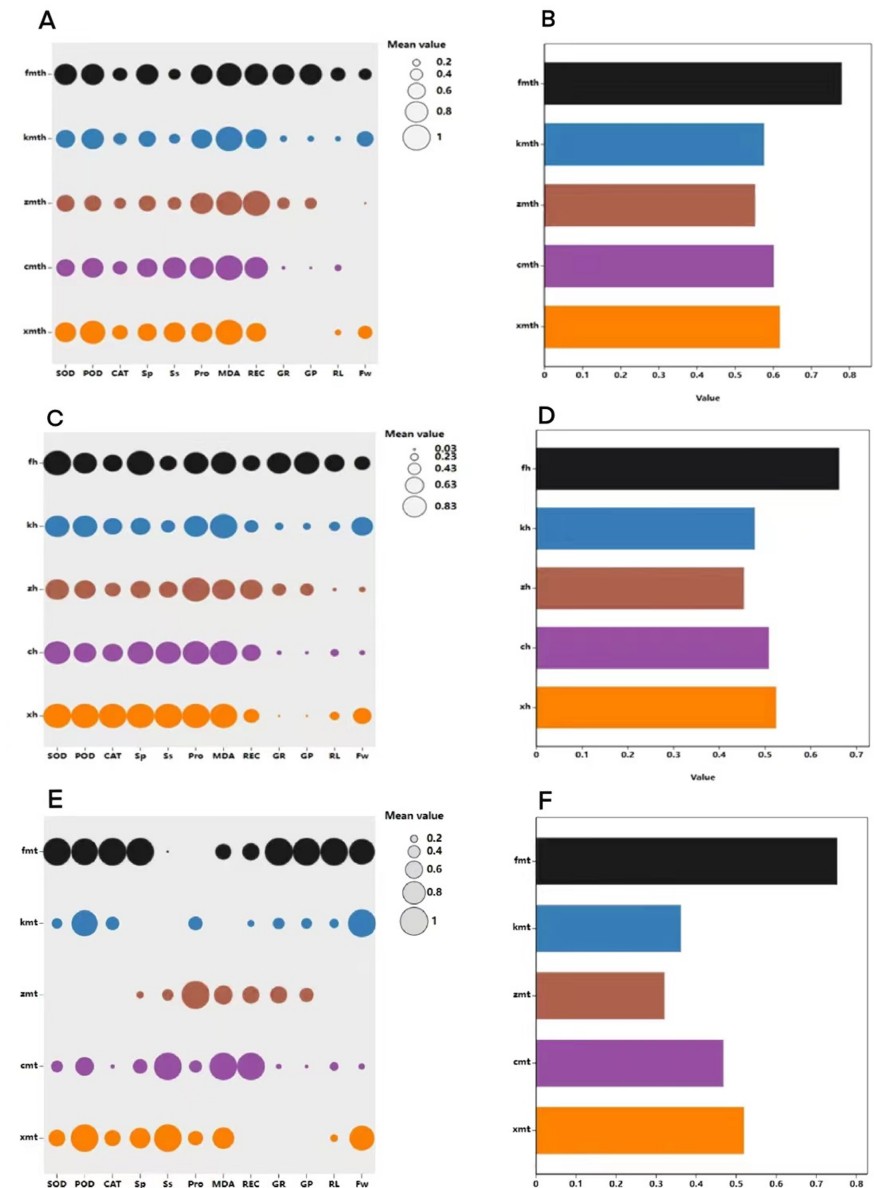

**Figure 13 Comprehensive evaluation of MT and high temperature stress on each index.**

# DISCUSSION

High-temperature stress markedly impairs seed germination and metabolism *via* diverse mechanisms. A seed reaction to such stress is evident from the suppression of its germination rate. The detrimental effects of high-temperature stress on seed germination chiefly manifest in the diminished germination rate and vigor, undermining the seed antioxidant defense and the synthesis of osmotic-adjustment compounds (*Hussain, Farooq & Lee, 2017*; *Shu et al., 2018*; *Sharma & Zheng, 2019*). Many studies have proven that high temperature stress significantly inhibits seed germination of chickpea genotypes
(*Adishesha & Chimmad, 2021*), rice (*Wang et al., 2022*), and beans (*De Santis et al., 2021*). In this research, five kinds of buckwheat seed growth indicators and germination were reduced by a high temperature of 28 °C—highlighting the adverse effects of high-temperature stress on buckwheat seed germination (Fig. 1). However, treatment with 200 μM MT under high-temperature conditions significantly enhanced the germination rate and potential, seed radicle growth, and fresh weight. These findings align well with prior research, which showed that an appropriate concentration of MT can bolster germination in the seed of rice, corn, wheat, and other seeds under stressful conditions, and stimulate radicle lengthening (*Kołodziejczyk, Kaźmierczak & Posmyk, 2021*; *Lei et al., 2021*; *Yu et al., 2022*). In this study, soaking buckwheat seeds with MT could alleviate the inhibition effect of high temperature on seed germination. Under the same high temperature stress treatment, seeds soaked with 200 μM MT had faster germination, higher radicle length, and higher fresh weight than seeds not treated with MT. In addition, 50 μM MT treatment had little effect on buckwheat seed germination (Table 2). The results showed that the effect of MT on buckwheat seed germination was closely related to its concentration, and 200 μM MT was the best concentration to promote buckwheat seed germination under high temperature stress. These findings are inconsistent with the results of 100 μM MT can promote the germination of cotton seed, which may be related to the variety.

High temperature stress during seed germination produces harmful substances, leading to oxidative damage in plants. In order to inhibit the excessive production of harmful substances, plants have a complex enzyme defense system to combat oxidative damage (*Ahammed et al., 2020*). Exogenous MT bolsters high-temperature tolerance primarily by eradicating reactive oxygen species (ROS) and amplifying antioxidant enzyme activity (*Qi et al., 2018*; *Jahan et al., 2021*). When exposed to oxidative stress from adverse conditions, plants form a suite of antioxidant enzymes within their cells, notably SOD, POD, and CAT (*Alam et al., 2018*). SOD scavenge superoxide anion free radicals, while POD and CAT $H_2O_2$ into $O_2$ and $H_2O$ further convert $H_2O_2$ into $O_2$ and $H_2O$ through oxidation and reduction, neutralizing toxic effects and facilitating seed germination (*Feng & Wang, 2020*). In this investigation, under high-temperature stress, the application of exogenous MT significantly elevated the activities of SOD, POD, and CAT across the five buckwheat varieties (Figs. 5, 6 and 7). This indicates that the antioxidant oxidase system in buckwheat seeds can clear ROS under high temperature stress. At the same time, this has been confirmed in *Aosophila alpini* (*Alam et al., 2018*) and *Pellinium ternata* (*Ma et al., 2020*). Among them, exogenous MT has the most significant effect on improving SOD activity in 'Fhth', 'Zk3h', and 'Cq8h'; however, exogenous MT has varying effects on the activity of POD and CAT at different germination times. This variation might be related to the varieties. Exogenous MT primarily regulates SOD activity in buckwheat seeds in the antioxidant enzyme system, while the activities of POD and CAT are influenced not just by exogenous MT but also by different varieties. This further indicates that while exogenous MT has varying impacts on the germination potential and rate of the five buckwheat seeds, all varieties regulate the activity of antioxidant enzymes to remove ROS, counteract the harmful effects of toxins, and foster seed germination.

The accumulation of harmful substances in seeds can induce lipid peroxidation and the degradation of unsaturated fatty acids, leading to the production of malondialdehyde (MDA), which subsequently causes structural damage to seeds and reduces their germination rate and potential (*Xiao et al., 2019*). Therefore, MDA is a key indicator of oxidative stress and can indicate lipid peroxidation in plant cell membranes (*Imran et al., 2021*). In this study, with the extension of seed treatment time, high temperature stress increased the lipid peroxidation damage of cell membrane and thus increased the MDA content, which was consistent with the results of *Li et al. (2022)*. However, treatment with 200 μM MT can effectively inhibit the accumulation of MDA in buckwheat seeds under high temperature stress (Fig. 8). This result reinforces the role of exogenous MT in enhancing peroxidase activity and improving cell membrane stability under high temperature stress, thereby mitigating membrane damage in buckwheat seeds. REC measures the extent of cell membrane damage in plants under stress. Studies have shown that stress can destroy the structure and function of the membrane, resulting in a significant increase in relative conductivity (*Xie et al., 2023*), which is consistent with the findings of this study. However, the introduction of 200 μM exogenous MT decreased the relative conductivity of five buckwheat seeds under high temperature stress (Table 4). This indicated that exogenous MT could enhance the membrane permeability of seed cells.

Seed germination is regulated by the presence of several storage products (*Fleming, Richards & Walters, 2017*). Osmotic adjustment substances, including Pro, Sp, and Ss, are pivotal in seed stress resistance. Under abiotic stress, seeds enhance their tolerance by accumulating Pro (*Yuan et al., 2022*). Pro is a crucial osmotic adjustment substance in plants, pivotal for maintaining cellular structural equilibrium and bolstering osmotic adjustment capabilities (*Gao et al., 2019*). Ss and Sp are energy sources for cells and important osmotic regulators, playing a protective role in cells (*Park et al., 2021*). High temperature stress significantly increased the Pro content in five buckwheat species, which shows that buckwheat can resist damage to cell structure caused by high temperature through the regulation of Pro under high temperature stress. And in this study, five kinds of buckwheat seeds were treated with 200 μM MT, which significantly increased the intraspecific Pro content under high temperature stress (Fig. 2). Highlighting MT's role in augmenting Pro levels in buckwheat seeds, thereby mitigating high-temperature-induced damage. Additionally, the application of exogenous MT significantly elevated the Ss and Sp contents in these buckwheat seeds, mirroring results observed by *Gao et al. (2018)*. This suggests that introducing an appropriate MT concentration can effectively raise the content of free Pro, Ss, and Sp during temperature stress, thus balancing intracellular osmotic potential and pressures, alleviating stress, and minimizing cellular membrane damage in seeds, thereby enhancing seed germination potential and rates.

In order to conduct a more comprehensive analysis, grey correlation analysis was employed to examine the effects of exogenous MT on the germination and growth indicators of buckwheat seeds under high-temperature stress. The results indicate that the activity of POD and Ss effectively reflect the relieving effect of exogenous MT on high temperature stress, compared with previous studies, which have suggested that the activity of POD and MDA content reflect the relieving effect of MT on waterlogging stress

(*Zeng et al., 2022*). The inconsistent results may be due to the fact that exogenous MT can induce specific physiological responses in plants under various abiotic stresses, thereby helping to resist the effects of these abiotic stresses. The study believes that exogenous MT regulates the antioxidant enzyme system to resist damage caused by abiotic stress (*Park et al., 2021*), which was consistent with our research findings. Through correlation analysis, it was found that there was a significant limit correlation ($p < 0.01$) among the activities of SOD, POD, and CAT. This indicates that exogenous MT reduces the damage to buckwheat seeds caused by high temperature stress by regulating the antioxidant enzyme system. However, through path coefficient analysis, it was found that exogenous MT has a significant impact on physiological indicators within buckwheat seeds under high temperature stress, with SOD activity being the most prominent, followed by Ss. In terms of indirectly regulating the membrane permeability of buckwheat seed cells, SOD activity also has the greatest impact, followed by Ss are important osmoregulation substances. Therefore, under high temperature stress conditions, exogenous MT can not only regulate the antioxidant enzyme system, but also regulate osmotic regulatory substances. However, the antioxidant enzyme system plays a leading role, and osmoregulation substances have a synergistic effect to resist the damage of high temperature stress to the cell membrane of buckwheat seeds. The permeability of the cell membrane is a key factor determining cell life and death, therefore, the protective effect of exogenous MT is crucial for the survival of buckwheat seeds under extreme temperatures. This provides us with a new perspective on understanding and utilizing exogenous MT to enhance seed germination under adverse conditions, especially in the context of global warming and an increase in high-temperature events.

Membership functions have been widely used for evaluating the resistance of varieties, and ranking the resistance of different types or varieties (*Narimani et al., 2011*). The salt tolerance of soybeans (*Bao et al., 2023*), and drought resistance evaluation of cotton have all been fully verified through membership functions (*Sun et al., 2021*). Our research results reveal that soaking buckwheat seeds with MT can significantly enhance their high-temperature tolerance. Comparing the average values of various membership functions, we found that 'Fhth' exhibited the best high-temperature resistance in high-temperature environments. In MT treatment, soaking seeds with 200 μM of MT had the most significant growth promoting effect on 'Fhth', with the highest membership function value of its related indicators. Further analysis of the treatment effects of high temperature and MT has confirmed that the average membership function of 'Fhth' is still the highest, demonstrating its excellent high-temperature resistance. This study delves into how exogenous MT can alleviate the adverse effects of high temperature stress on buckwheat seed germination at the physiological level, providing us with a new strategy to enhance seed high-temperature tolerance *via* exogenous MT.

## CONCLUSIONS

This study evaluated the efficacy of MT treatment in enhancing the GR, GP, FW, SOD, POD, CAT and decreasing the contents of MDA of buckwheat seeds under high temperature

stress conditions. Through the path coefficient analysis, the promotion effect of MT on seed germination may be mainly related to the antioxidant enzyme system. To conclude, exogenous application of MT effectively augmented the germination capability of five buckwheat seeds under high temperature stress. This study also elucidated the potential mechanism underlying MT's role in enhancing high temperature endurance of buckwheat seeds, grounded in their morphological and physiological characteristics.

## ACKNOWLEDGEMENTS

The authors are grateful to the anonymous reviewers for their valuable comments and suggestions.

### Funding

This work was supported by the National Natural Science Foundation of China (No. 32372045), the Science and Technology Project of Hebei Education Department (BJ2019022) and the Natural Science Foundation of Hebei Province (C2023204097). The funders had no role in study design, data collection and analysis, decision to publish, or preparation of the manuscript.

### Grant Disclosures

The following grant information was disclosed by the authors:
National Natural Science Foundation of China: 32372045.
Science and Technology Project of Hebei Education Department: BJ2019022.
Natural Science Foundation of Hebei Province: C2023204097.

### Competing Interests

The authors declare there are no competing interests.

### Author Contributions

- Zemiao Tian conceived and designed the experiments, performed the experiments, authored or reviewed drafts of the article, and approved the final draft.
- Mengyu Zhao conceived and designed the experiments, performed the experiments, authored or reviewed drafts of the article, and approved the final draft.
- Junzhen Wang analyzed the data, prepared figures and/or tables, and approved the final draft.
- Qian Yang performed the experiments, prepared figures and/or tables, and approved the final draft.
- Yini Ma performed the experiments, prepared figures and/or tables, and approved the final draft.
- Xinlei Yang analyzed the data, prepared figures and/or tables, and approved the final draft.
- Luping Ma analyzed the data, prepared figures and/or tables, and approved the final draft.

- Yongzhi Qi analyzed the data, prepared figures and/or tables, and approved the final draft.
- Jinbo Li analyzed the data, prepared figures and/or tables, and approved the final draft.
- Muriel Quinet conceived and designed the experiments, authored or reviewed drafts of the article, and approved the final draft.
- BaoSheng Shi conceived and designed the experiments, authored or reviewed drafts of the article, and approved the final draft.
- Yu Meng conceived and designed the experiments, authored or reviewed drafts of the article, and approved the final draft.

## Data Availability

The raw measurements are available in the Supplementary File.

## Supplemental Information

Supplemental information for this article can be found online at http://dx.doi.org/10.7717/peerj.17136#supplemental-information.

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
