# Peer review of "Exogenous melatonin improves germination rate in buckwheat under high temperature stress by regulating seed physiological and biochemical characteristics"

_PeerJ, doi:10.7717/peerj.17136_

## Round 0.1 · original submission · Major Revisions

Dear authors,

Three reviewers have now submitted their assessment of your manuscript. Several concerns have been raised about what seems to be a significant drawback, the novelty of the study. Hence, I would suggest making a more precise description of the novel findings of your study in the revised manuscript.

Sincerely, dr Nikolaos Nikoloudakis

·

Basic reporting

Basic reporting
The manuscript lacks the professional and unambiguous language required for scientific reporting. The text needs substantial revisions to enhance clarity, coherence, and adherence to academic language standards. The writing style should be refined to meet the expected standards of the journal.
The manuscript provides context but lacks depth in linking the research to existing literature. While some references are included, the background information does not sufficiently highlight the gap in knowledge that this research aims to fill. A more comprehensive review of relevant literature is needed to establish the significance of the study.
The figures provided lack adequate description and labeling, affecting their overall quality. Additionally, the raw data supporting the findings have not been supplied as per the journal's policy. The authors should include high-quality, relevant figures with detailed descriptions and provide raw data to support their conclusions.
Recommendations:
Language Enhancement: The authors should revise the manuscript to ensure professional, clear, and precise language throughout the document.
Literature Review Improvement: Strengthen the background by incorporating more relevant literature and explicitly stating how this research addresses the identified gap in knowledge.
Figure Quality and Raw Data Submission: Improve figure quality, ensure proper labeling and description, and provide the raw data to comply with the journal's policy.
Methodological Detail: Enhance the description of methods to allow for effective replication by other researchers.
1. Italicize scientific names, in line 22
2. Start new paragraph with a new sentence. Don’t loose the continuity of a passage. (Line 83-85).
3. Use unit with last value only, always add space between value and unit used.
4. Use only SI units.
5. Follow abbreviation rules, use abbreviation with its full form first and then only use abbreviated form only.
6. Remove extra spacing (120-121). Check whole manuscript for this.
7. You may use % symbol with last value only in the results.
8. Reference styles are not uniform and do not match journal’s guideline. Kindly use same reference style.

Experimental design

Experimental design
The research question is well defined and relevant. However, the manuscript lacks clarity in explaining how this study fills the identified knowledge gap. A more explicit connection between the research question and the gap in the existing literature should be established.

Validity of the findings

Validity of Findings
The manuscript presents findings that are supported by robust data and statistical analyses, contributing to the overall validity of the study. The data presented are consistent with the research objectives and are controlled appropriately, enhancing the reliability of the conclusions drawn.

Additional comments

The figures provided lack adequate description and labeling, affecting their overall quality. Additionally, the raw data supporting the findings have not been supplied as per the journal's policy. The authors should include high-quality, relevant figures with detailed descriptions and provide raw data to support their conclusions.

Reviewer 2 ·

Basic reporting

I congratulate all the authors for this excellent study. However there are some sections that needs revisions.

1: Remove the abbreviations used in the abstract.
2. Botanical names must be italicized and checked properly

3: Introduction is well managed but needs little improvement regarding study gap, hypothesis, and clear objectives. Novelty of the study should be mentioned at the end of introduction section.
4: """, soaked them in 75% alcohol and 10% sodium hypochlorite for 10 minutes, then rinsed and dried them with distilled.? Reference of the protocol used be added
5 : 280°C... Correct it.
6: experimental set-up should be added separately


7: methodology section need revision and should be rewritten in detailed manner.
8: Try to discuss results with recent literature and provide reasoning of the responses recorded. Improve the discussion with logical and reasoning approaches.
8: Conclusion must be short, specific, and quantified. Add novel findings of your study .

Apart from this there are various. Grammatical and language errors, that need to be corrected in the revised versions

Hope my suggestions would improve the quality of ms. Thank you

Experimental design

Methods should be described with sufficient detail & information to replicate

Validity of the findings

No comments
Try to discuss results with recent literature and provide reasoning of the responses recorded. Improve the discussion with logical and reasoning approaches.
Conclusion must be short, specific, and quantified. Add novel findings of your study .

Additional comments

No comments

Reviewer 3 ·

Basic reporting

I have gone through the manuscript and found that “Exogenous melatonin improves germination rate in buckwheat under high temperature stress by regulating seed physiological and biochemical characteristics”. The author studied the impacts of melatonin seed soaking and high-temperature stress treatments on the morphology and physiology potential of melatonin seed soaking for the improving germination of buckwheat under temperature stress conditions. The manuscript has a good application value. However, the novelty of this work was poor to meet the standard of publication in the journal. So, it is hard for me to support publish this study in the journal. Also too many problem and English language is needed that the reviewers will easily understand your meaning. I suggest authors to submit this manuscript to other relevant agronomy journals.

Experimental design

Why were these genotypes selected for the experiment? Please add the full information and genetic background of these genotypes.
Where are your measurement methods? Please write a detailed method of measurement indicators.

Validity of the findings

. The manuscript has a good application value. However, the novelty of this work was poor to meet the standard of publication in the journal. So, it is hard for me to support publish this study in the journal. Also too many problem and English language is needed that the reviewers will easily understand your meaning. I suggest authors to submit this manuscript to other relevant agronomy journals.

Additional comments

Several amendments are suggested as follows:
1. The hypothesis is not clear.
2. Did the authors have checked the genetic background of the used genotypes?
3. Please provided detailed information on the number of seedlings per replicate and number of replicates, to improve the understandability of the experimental setup and the statistical analyses.
4. The abstract does not summarize the results of the article.
5. Improve the whole abstract.
6. Why these genotypes were selected for the experiment. Please add the full information and genetic background of these genotypes.
10. Where is your measurement methods? Please write a detail method of measurements indicators?
12. Language needs substantial improvement.
16. Discussion is not good to meet the quality of the journal need more justification and explanation?
17. Revise the whole manuscript?
18. Carefully revise all the figure legends for grammatical, spelling, and spacing mistakes.

---

## Round 0.2 · Minor Revisions

Dear Dr. Tian, the experts re-evaluated your manuscript and advised acceptance after minor revisions.

best
Dr Nikolaos Nikoloudakis

·

Basic reporting

Authors have revised the manuscript as per suggestions of the reviewers. Therefore, I recommend that manuscript can be accepted for publication in Peer J.

Experimental design

Authors have revised the manuscript as per suggestions of the reviewers. Therefore, I recommend that manuscript can be accepted for publication in Peer J.

Validity of the findings

Authors have revised the manuscript as per suggestions of the reviewers. Therefore, I recommend that manuscript can be accepted for publication in Peer J.

Additional comments

Authors have revised the manuscript as per suggestions of the reviewers. Therefore, I recommend that manuscript can be accepted for publication in Peer J.

Reviewer 2 ·

Basic reporting

I have few queries before accepting this article

1: the abstract section need to be improved.
2 novelty, research gap and highlights should be given at the end of introduction section
3: language should be improved to maintain the smoothnes of the ms
4 rewrite the conclusion section. Add the main findings of your study

Experimental design

Need to be given in brief

Validity of the findings

Fine

Additional comments

No

---

## Round 0.3 · accepted · Accept

the authors have addressed all of the reviewers' comments, i believe the MS can be accepted for publication

In addition, the Section Editor commented:

LINE NO: / BEFORE / AFTER / [COMMENTS]
LINE 74: / . / . / [not clear in what sense the term ‘Mosaic’ is being used here; clarify.]